# Approximating Two-Layer ReLU Networks for Hidden State Analysis in Differential Privacy

## Abstract

The hidden state threat model of differential privacy (DP) assumes that the adversary has access only to the final trained machine learning (ML) model, without seeing intermediate states during training. Current privacy analyses under this model, however, are limited to convex optimization problems, reducing their applicability to multi-layer neural networks, which are essential in modern deep learning applications. Additionally, the most successful applications of the hidden state privacy analyses in classification tasks have been for logistic regression models. We demonstrate that it is possible to privately train convex problems with privacy-utility trade-offs comparable to those of one hidden-layer ReLU networks trained with DP stochastic gradient descent (DP-SGD). We achieve this through a stochastic approximation of a dual formulation of the ReLU minimization problem which results in a strongly convex problem. This enables the use of existing hidden state privacy analyses, providing accurate privacy bounds also for the noisy cyclic mini-batch gradient descent (NoisyCGD) method with fixed disjoint mini-batches. Our experiments on benchmark classification tasks show that NoisyCGD can achieve privacy-utility trade-offs comparable to DP-SGD applied to one-hidden-layer ReLU networks. Additionally, we provide theoretical utility bounds that highlight the speed-ups gained through the convex approximation.

## 1 Introduction

In differentially private (DP) machine learning (ML), the DP-SGD algorithm (see e.g., Abadi et al., 2016) has become a standard tool obtain ML models with strong privacy guarantees of the individuals. The $(\varepsilon, \delta)$-guarantees for DP-SGD are obtained by clipping gradients by adding normally distributed noise scaled with the clipping constant to the randomly sampled mini-batch of gradients, and by using composition DP analysis (see, e.g., Zhu et al., 2022).

One weak point of the composition analysis of DP-SGD is that it is assumes that the adversary has access to all the intermediate results of the training iteration. This assumption is often unnecessarily strict as in many practical scenarios only the final model is needs to be revealed. Another weakness is that it requires either full batch training or random subsampling, and, e.g., accurate privacy analyses for of many practically relevant algorithms are not available for non-convex problems such as for training multi-layer neural networks. One example of such algorithms is the noisy cyclic GD (NoisyCGD) with disjoint mini-batches. Having high privacy-utility ML models trained with NoisyCGD would give an alternative for DP-SGD that is often difficult to implement in practical settings (Chua et al., 2024a).

The so called hidden state threat model of DP considers releasing only the final model of the training iteration, and existing $(\varepsilon, \delta)$-DP analyses in the literature are only applicable for convex problems such as the logistic regression which is the highest performing model considered in the literature (see, e.g., Chourasia et al., 2021; Bok et al., 2024). When training models with DP-SGD, however, one quickly finds that the model performance of commonly used convex models is inferior compared to multi-layer neural networks. A natural question then arises whether convex approximations of minimization problems for multi-layer neural networks can be made while preserving model performance. In response, this work explores such an approximation for the two-layer ReLU minimization problem. To achieve this, we build on the findings of Pilanci &

Ergen (2020), which demonstrate the existence of a convex dual formulation for the two-layer ReLU minimization problem when the hidden layer is sufficiently wide.

The privacy amplification by iteration analysis for convex private optimization, introduced by Feldman et al. (2018), provides privacy guarantees in the hidden state threat model. However, this and many subsequent analyses (Sordello et al., 2021; Asoodeh et al., 2020; Chourasia et al., 2021; Altschuler & Talwar, 2022) remain challenging to apply in practice, as it typically requires a large number of training iterations for obtaining tighter DP guarantees than those of DP-SGD. Chourasia et al. (2021) improved this analysis using Rényi DP for full-batch DP-GD training, while Ye & Shokri (2022) offered a similar analysis for shuffled mini-batch DP-SGD. Recently, Bok et al. (2024) provided an $f$-DP analysis for a class of algorithms, which we also leverage to analyze NoisyCGD.

Also from a theoretical standpoint, convex models are advantageous over non-convex ones in private optimization. State-of-the-art empirical risk minimization (ERM) bounds for private convex optimization are of the order $O(\frac{1}{\sqrt{n}} + \frac{\sqrt{d}}{\varepsilon n})$, where $n$ is the number of training data entries, $d$ the dimension of the parameter space and $\varepsilon$ the DP parameter (Bassily et al., 2019). In contrast, the bounds for non-convex optimization, which focus on finding stability points, are much worse, such as $O(\frac{1}{n^{1/3}} + \frac{d^{1/5}}{(\varepsilon n)^{2/5}})$ (Bassily et al., 2021) and $O\left(\frac{d^{1/3}}{(\varepsilon n)^{2/3}}\right)$ (Arora et al., 2023; Lowy et al., 2024).

Our main contributions are the following:

- By integrating two seemingly unrelated approaches, convex reformulation of ReLU networks and privacy amplification by iteration DP analysis, we show that it is possible to obtain similar privacy-utility trade-offs in the hidden state threat model of DP as by applying DP-SGD to two-layer ReLU networks and using composition results.
- We carry out a number of approximations for the convex reformulation to facilitate DP analysis and show that the resulting strongly convex model has the required properties for hidden state analysis.
- We give the first high privacy-utility trade-off results for benchmark classification tasks using a hidden state DP analysis. In particular, we give the first empirical high privacy-utility trade-off results for NoisyCGD with disjoint mini-batches under the hidden state threat model which makes it more suitable for practical applications of DP ML. The experiments also account for the privacy cost of hyperparameter tuning, and we demonstrate how to conduct it effectively for NoisyCGD.
- We carry out a theoretical utility analysis of DP-SGD applied to the convex approximation within the random data model.

## 2 PRELIMINARIES

We denote a dataset containing $n$ data points as $D = (z_1, \ldots, z_n)$. We say $D$ and $D'$ are neighboring datasets if they differ in exactly one element (denoted as $D \sim D'$). We say that a mechanism $\mathcal{M} : \mathcal{X} \to \mathcal{O}$ is $(\varepsilon, \delta)$-DP if the output distributions for neighboring datasets are always $(\varepsilon, \delta)$-indistinguishable (Dwork et al., 2006).

**Definition 2.1.** Let $\varepsilon \geq 0$ and $\delta \in [0, 1]$. Mechanism $\mathcal{M} : \mathcal{X} \to \mathcal{O}$ is $(\varepsilon, \delta)$-DP if for every pair of neighboring datasets $D, D'$ and for every measurable set $E \subset \mathcal{O}$,

$$\mathbb{P}(\mathcal{M}(D) \in E) \leq \mathrm{e}^\varepsilon \mathbb{P}(\mathcal{M}(D') \in E) + \delta.$$

We call $\mathcal{M}$ tightly $(\varepsilon, \delta)$-DP, if there does not exist $\delta' < \delta$ such that $\mathcal{M}$ is $(\varepsilon, \delta')$-DP.

**Hockey-stick Divergence and Numerical Privacy Accounting.** The DP guarantees can be alternatively described using the hockey-stick divergence which is defined as follows. For $\alpha > 0$ the hockey-stick divergence $H_\alpha$ from a distribution $P$ to a distribution $Q$ is defined as

$$H_\alpha(P||Q) = \int [P(t) - \alpha \cdot Q(t)]_+ \, \mathrm{d}t, \tag{2.1}$$

where for $t \in \mathbb{R}$, $[t]_+ = \max\{0, t\}$. The $(\varepsilon, \delta)$-DP guarantee as defined in Def. 2.1 can be characterized using the hockey-stick divergence as follows.

**Lemma 2.2** (Zhu et al. 2022). *For a given $\varepsilon \geq 0$, tight $\delta(\varepsilon)$ is given by the expression*

$$\delta(\varepsilon) = \max_{D \sim D'} H_{e^{\varepsilon}}(\mathcal{M}(D)||\mathcal{M}(D')).$$

Thus, if we can bound the divergence $H_{e^{\varepsilon}}(\mathcal{M}(D)||\mathcal{M}(D'))$ accurately, we also obtain accurate $\delta(\varepsilon)$-bounds. We also refer to $\delta_{\mathcal{M}}(\varepsilon) := \max_{D \sim D'} H_{e^{\varepsilon}}(\mathcal{M}(D)||\mathcal{M}(D'))$ as the *privacy profile* of mechanism $\mathcal{M}$. For bounding the hockey-stick divergence of compositions accurately, we need to so-called dominating pairs of distributions.

**Definition 2.3** (Zhu et al. 2022). A pair of distributions $(P, Q)$ is a *dominating pair* of distributions for mechanism $\mathcal{M}(D)$ if for all neighboring datasets $D$ and $D'$ and for all $\alpha > 0$,

$$H_{\alpha}(\mathcal{M}(D)||\mathcal{M}(D')) \leq H_{\alpha}(P||Q).$$

If the equality holds for all $\alpha$ for some $D, D'$, then $(P, Q)$ is a tightly dominating pair of distributions. We get upper bounds for DP-SGD compositions using the dominating pairs of distributions using the following composition result.

**Theorem 2.4** (Zhu et al. 2022). *If $(P, Q)$ dominates $\mathcal{M}$ and $(P', Q')$ dominates $\mathcal{M}'$, then $(P \times P', Q \times Q')$ dominates the adaptive composition $\mathcal{M} \circ \mathcal{M}'$.*

To convert the hockey-stick divergence from $P \times P'$ to $Q \times Q'$ into an efficiently computable form, we consider so called privacy loss random variables (PRVs) and use Fast Fourier Technique-based methods (Koskela et al., 2021; Gopi et al., 2021) to numerically evaluate the convolutions appearing when summing the PRVs and evaluating $\delta(\varepsilon)$ for the compositions.

**Gaussian Differential Privacy.** For the privacy accounting of the noisy cyclic mini-batch GD, we use the bounds by Bok et al. (2024) that are stated using the Gaussian differential privacy (GDP). Informally speaking, a mechanism $\mathcal{M}$ is $\mu$-GDP, $\mu \geq 0$, if for all neighboring datasets the outcomes of $\mathcal{M}$ are not more distinguishable than two unit-variance Gaussians $\mu$ apart from each other (Dong et al., 2022). We consider the following formal characterization of GDP.

**Lemma 2.5** (Dong et al. 2022, Cor. 2.13). *A mechanism $\mathcal{M}$ is $\mu$-GDP if and only it is $(\varepsilon, \delta)$-DP for all $\varepsilon \geq 0$, where*

$$\delta(\varepsilon) = \Phi\left(-\frac{\varepsilon}{\mu} + \frac{\mu}{2}\right) - e^{\varepsilon}\Phi\left(-\frac{\varepsilon}{\mu} - \frac{\mu}{2}\right).$$

## 2.1 DP-SGD with Poisson Subsampling

DP-SGD iteration with Poisson subsampling is given by

$$\theta_{j+1} = \theta_j - \eta_j \cdot \left(\frac{1}{b}\sum\nolimits_{x \in B_j} \text{clip}(\nabla\mathcal{L}(x, \theta_j), C) + Z_j\right), \tag{2.2}$$

where $C > 0$ denotes the clipping constant, $\text{clip}(\cdot, C)$ the clipping function that clips gradients to have 2-norm at most $C$, $\mathcal{L}$ the loss function, $\theta$ the model parameters, $\eta_j$ the learning rate at iteration $j$, $B_j$ the mini-batch at iteration $j$ that is sampled with Poisson subsampling with the subsampling ratio $b/n$, $b$ the expected size of each mini-batch and $Z_j \sim \mathcal{N}(0, \frac{C^2\sigma^2}{b^2}I_d)$ the noise vector.

We want to experimentally compare DP-SGD to the noisy cyclic mini-batch gradient descent using the privacy amplification by iteration analysis Bok et al. (2024). To this end, we consider the substitute neighborhood relation of datasets. To this end, we use to following results by Lebeda et al. (2024).

**Lemma 2.6** (Lebeda et al. 2024). *Suppose a pair of distributions $(P, Q)$ is a dominating pair of distributions for a mechanism $\mathcal{M}$ and denote the Poisson subsampled mechanism $\widetilde{\mathcal{M}} := \mathcal{M} \circ S^q_{Poisson}$, where $S^q_{Poisson}$ denotes the Poisson subsampling with subsampling ratio $q$. Then, for all neighbouring datasets (under the $\sim$-neighbouring relation) $D$ and $D'$,*

$$\begin{aligned}
&H_{\alpha}\big(\widetilde{\mathcal{M}}(D)||\widetilde{\mathcal{M}}(D')\big) \\
&\leq H_{\alpha}\big((1-q)\cdot\mathcal{N}(0, \sigma^2) + q\cdot\mathcal{N}(1, \sigma^2)||(1-q)\cdot\mathcal{N}(0, \sigma^2) + q\cdot\mathcal{N}(-1, \sigma^2)\big)
\end{aligned} \tag{2.3}$$

*for all $\alpha \geq 0$.*

Furthermore, using the composition result of Lemma 2.4 and numerical accountants, we obtain $(\varepsilon, \delta)$-bounds for compositions of DP-SGD with Poisson subsampling the substitute neighborhood relation of datasets. Alternatively, we could use RDP bounds given by Wang et al. (2019), however, as also illustrated by the Appendix Figure 6, our numerical approach generally leads to tighter bounds.

## 2.2 Guarantees for the Final Model and for Noisy Cyclic Mini-Batch GD

We next consider privacy amplification by iteration (Feldman et al., 2018) type of analysis that gives DP guarantees for the final model of the training iteration. We use the recent results by Bok et al. (2024) that are applicable to the noisy cyclic mini-batch gradient descent (NoisyCGD) for which one epoch of training is described by the iteration

$$\theta_{j+1} = \theta_j - \eta \left( \frac{1}{b} \sum_{x \in B_j} \nabla_\theta f(\theta_j, x) + Z_j \right) \tag{2.4}$$

where $Z_j \sim \mathcal{N}(0, \sigma^2 I_d)$ and the data $D$, $|D| = n$, is divided into disjoint batches $B_1, \ldots, B_k$, each of size $b$. The analysis by Bok et al. (2024) considers the substitute neighborhood relation of datasets and central for the DP analysis is the gradient sensitivity.

**Definition 2.7.** We say that a family of loss functions $\mathcal{F}$ has a gradient sensitivity $L$ if

$$\sup_{f,g \in \mathcal{F}} \|\nabla f - \nabla g\| \leq L.$$

As an example relevant to our analysis, for a family of loss functions of the form $h_i + r$, where $h_i$'s are $L$-Lipschitz loss functions and $r$ is a regularization function, the sensitivity equals $2L$. We will use the following result for analysing the $(\varepsilon, \delta)$-DP guarantees of NoisyCGD. First, recall that a function $f$ is $\beta$-smooth if $\nabla f$ is $\beta$-Lipschitz, and it is $\lambda$-strongly convex if the function $g(x) = f(x) - \frac{\lambda}{2} \|x\|_2^2$ is convex.

**Theorem 2.8** (Bok et al. 2024, Thm. 4.5). *Consider $\lambda$-strongly convex, $\beta$-smooth loss functions with gradient sensitivity $L$. Then, for any $\eta \in (0, 2/M)$, NoisyCGD is $\mu$-GDP for*

$$\mu = \frac{L}{b\sigma} \sqrt{1 + c^{2k-2} \frac{1 - c^2}{(1 - c^k)^2} \frac{1 - c^{k(E-1)}}{1 + c^{k(E-1)}}},$$

*where $k = n/b$, $c = \max\{|1 - \eta\lambda|, |1 - \eta\beta|\}$ and $E$ denotes the number of epochs.*

We could alternatively use the RDP analysis by Ye & Shokri (2022), however, as also illustrated by the experiments of Bok et al. (2024), the bounds given by Thm. 2.8 lead to slightly lower $(\varepsilon, \delta)$-DP bounds for NoisyCGD.

In order to benefit from the privacy analysis of Thm. 2.8 for NoisyCGD, we add an $L_2$-regularization term with a coefficient $\frac{\lambda}{2}$ which makes the loss function $\lambda$-strongly convex. Finding suitable values for the learning rate $\eta$ and regularization parameter $\lambda$ is complicated by the following aspects. The larger the regularization parameter $\lambda$ and the learning rate $\eta$ are, the faster the model 'forgets' the past updates and the faster the $\varepsilon$-values converge. This is reflected in the GDP bound of Thm. 2.8 in the constant $c$ which generally equals $|1 - \eta\lambda|$. Thus, in order to benefit from the bound of Thm. 2.8, the product $\eta\lambda$ should not be too small. On the other hand, when $\eta\lambda$ is too large, the 'forgetting' starts to affect the model performance. We experimentally observe that the plateauing of the model accuracy and privacy guarantees happens approximately at the same time.

Figure 1 illustrates privacy guarantees of NoisyCGD for a range of values for the product $\eta\lambda$ where the $(\varepsilon, \delta)$-DP guarantees given by Thm. 2.8 become smaller than those given by the Poisson subsampled DP-SGD with an equal batch size $b = 1000$ when $\sigma = 15.0$ when training for 400 epochs. For a given value of the learning $\eta$, we can always adjust the value of $\lambda$ to have desirable $(\varepsilon, \delta)$-DP guarantees. To put the values of Fig. 1 into perspective, in experiments we observe that $\eta\lambda = 2 \cdot 10^{-4}$ is experimentally found to already affect the model performance considerably whereas $\eta\lambda = 1 \cdot 10^{-4}$ affects only weakly.

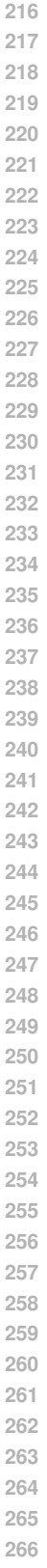

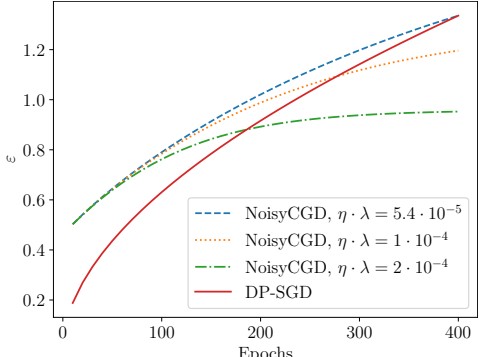

Figure 1: Values of the product of the learning rate $\eta$ and the $L_2$-regularization constant $\lambda$ that lead to tighter privacy bounds for the final model using Thm. 2.8 than for the whole sequence of updates using the DP-SGD analysis. Here the $\varepsilon$-values are shown as a function of number of training epochs, when total number of training samples $N = 6 \cdot 10^4$, the batch size $b = 1000$, $\sigma = 15.0$ and $\delta = 10^{-5}$.

## 3 CONVEX APPROXIMATION OF TWO-LAYER ReLU NETWORKS

We next derive step by step the strongly convex approximation of the 2-layer ReLU minimization problem and show that the derived problem is amenable for the privacy amplification by iteration type of $(\varepsilon, \delta)$-DP analysis. To simplify the presentation, we consider a 1-dimensional output network (e.g., a binary classifier). It will be straightforward to construct multivariate output networks from the scalar networks (see also Ergen et al., 2023a).

### 3.1 CONVEX DUALITY OF TWO-LAYER ReLU PROBLEM

We first consider the convex reformulation of the 2-layer ReLU minimization problem as presented by Pilanci and Ergen (2020). In particular, consider training a 2-layer ReLU network (with hidden-width $m$) $f : \mathbb{R}^d \to \mathbb{R}$,

$$f(x) = \sum_{j=1}^{m} \phi(u_j^T x) \alpha_j, \tag{3.1}$$

where the weights $u_i \in \mathbb{R}^d$, $i \in [m]$ and $\alpha \in \mathbb{R}^m$, and $\phi$ is the ReLU activation function, i.e., $\phi(t) = \max\{0, t\}$. For a vector $x$, $\phi$ is applied element-wise, i.e. $\phi(x)_i = \phi(x_i)$.

Using the squared loss and $L_2$-regularization with a regularization constant $\lambda > 0$, the 2-layer ReLU minimization problem can then be written as

$$\min_{\{u_i, \alpha_i\}_{i=1}^m} \frac{1}{2} \left\| \sum_{i=1}^{m} \phi(Xu_i)\alpha_i - y \right\|_2^2 + \frac{\lambda}{2} \sum_{i=1}^{m} (\|u_i\|_2^2 + \alpha_i^2), \tag{3.2}$$

where $X \in \mathbb{R}^{n \times d}$ denotes the matrix of the feature vectors, i.e., $X^T = \begin{bmatrix} x_1 & \dots & x_n \end{bmatrix}$ and $y \in \mathbb{R}^n$ denotes the vector of labels.

The convex reformulation of this problem is based on enumerating all the possible activation patterns of $\phi(Xu)$, $u \in \mathbb{R}^d$. The set of activation patterns that a ReLU output $\phi(Xu)$ can take for a data feature matrix $X \in \mathbb{R}^{n \times d}$ is described by the set of diagonal boolean matrices

$$\mathcal{D}_X = \{D = \mathrm{diag}(\mathbb{1}(Xu \geq 0)) : u \in \mathbb{R}^d\},$$

where for $i \in [n]$, $\big(\mathbb{1}(Xu \geq 0)\big)_i = 1$, if $(Xu)_i \geq 0$ and 0 otherwise. Here $|\mathcal{D}_X|$ is the number of regions in a partition of $\mathbb{R}^d$ by hyperplanes that pass through origin and are perpendicular to the rows of $X$. We have (Pilanci & Ergen, 2020):

$$|\mathcal{D}_X| \leq 2r \left( \frac{e(n-1)}{r} \right)^r,$$

where $r = \mathrm{rank}(X)$.

Let $|\mathcal{D}_X| = M$ and denote $\mathcal{D}_X = \{D_1, \ldots, D_M\}$. Let $\lambda > 0$. Next, the parameter space is partitioned into convex cones $C_1, \ldots, C_M$, $C_i = \{u \in \mathbb{R}^d : (2D_i - I)Xu \geq 0\}$, and we consider a convex optimization problem with group $\ell_2$ - $\ell_1$ - regularization

$$\min_{v_i, w_i} \frac{1}{2} \left\| \sum_{i \in [M]} D_i X(v_i - w_i) - y \right\|_2^2 + \lambda \sum_{i \in [M]} (\|v_i\|_2 + \|w_i\|_2) \tag{3.3}$$

such that for all $i$, $1 \leq i \leq M$ : $v_i, w_i \in C_i$ i.e.

$$(2D_i - I)Xw_i \geq 0, \quad (2D_i - I)Xv_i \geq 0.$$

Interestingly, for a sufficiently large hidden-width $m$, the ReLU minimization problem (3.2) and the convex problem (3.3) have equal minima.

**Theorem 3.1** (Pilanci & Ergen 2020, Thm. 1). *There exists $m^* \in \mathbb{N}$, $m^* \leq d + 1$, such that for all $m \geq m^*$, the ReLU minimization problem* (3.2) *and the convex problem* (3.3) *have equal minima.*

Moreover, Pilanci & Ergen (2020) show that for a large enough hidden-width $m$, the optimal weights of the ReLU network can be constructed from the optimal solution of the convex problem 3.3. Subsequent work, such as (Mishkin et al., 2022), extends the equivalence in Thm. 3.1 to general convex loss functions $\mathcal{L}$, rather than focusing solely on the squared loss. For simplicity, we focus on the squared loss in our presentation. We remark that convex formulations have also be shown for two-layer convolutional networks (Bartan & Pilanci, 2019) and for multi-layer ReLU networks (Ergen & Pilanci, 2021).

### 3.2 STOCHASTIC APPROXIMATION

Since $|\mathcal{D}_X|$ is generally an enormous number, stochastic approximations to the problem (3.3) have been considered (Pilanci & Ergen, 2020; Wang et al., 2022; Mishkin et al., 2022; Kim & Pilanci, 2024). In this approximation, vectors $u_i \sim \mathcal{N}(0, I_d)$, $i \in [P]$, $P \ll M$, are sampled randomly to construct the boolean diagonal matrices $D_1, \ldots, D_P$, $D_i = \text{diag}(\mathbb{1}(Xu_i \geq 0))$, and the problem (3.3) is replaced by

$$\min_{v_i, w_i} \frac{1}{2} \left\| \sum_{i=1}^{P} D_i X(v_i - w_i), y \right\|_2^2 + \lambda \sum_{i=1}^{P} (\|v_i\|_2 + \|w_i\|_2) \tag{3.4}$$

such that for all $i \in [P]$ : $v_i, w_i \in C_i$, i.e.,

$$(2D_i - I)Xw_i \geq 0, \qquad (2D_i - I)Xv_i \geq 0. \tag{3.5}$$

For practical purposes we consider a stochastic approximation of this kind. However, the constraints 3.5 are data-dependent which potentially makes private learning of the problem (3.4) difficult. Moreover, the overall loss function given by Eq. 3.4 is not generally strongly convex which prevents us using privacy amplification results such as Theorem 2.8 for NoisyCGD. We next consider a strongly convex problem without constraints of the form 3.5.

### 3.3 STOCHASTIC STRONGLY CONVEX APPROXIMATION

Motivated by experimental observations and also the formulation given in (Wang et al., 2022), we consider global minimization of the loss function (denote $v = \{v_i\}_{i=1}^{P}$)

$$\mathcal{L}(v, X, y) = \frac{1}{2n} \left\| \sum_{i=1}^{P} D_i X v_i - y \right\|_2^2 + \frac{\lambda}{2} \sum_{i=1}^{P} \|v_i\|_2^2, \tag{3.6}$$

where the diagonal boolean matrices $D_1, \ldots, D_P \in \mathbb{R}^{n \times n}$ are constructed by taking first $P$ i.i.d. samples $u_1, \ldots, u_P$, $u_i \sim \mathcal{N}(0, I_d)$, and then setting the diagonal elements of $D_i$'s as above as

$$(D_i)_{jj} = \max\left(0, \text{sign}(x_j^T u_i)\right).$$

Note that we may also write the loss function of Eq. (3.6) in the summative form

$$\mathcal{L}(v, X, y) = \frac{1}{n} \sum_{j=1}^{n} \ell(v, x_j, y_j), \tag{3.7}$$

where

$$\ell(v, x_j, y_j) = \frac{1}{2} \left\| \sum_{i=1}^{P} (D_i)_{jj} x_j^T v_i - y_j \right\|_2^2 + \frac{\lambda}{2} \sum_{i=1}^{P} \|v_i\|_2^2, \quad (D_i)_{jj} = \mathbb{1}(x_j^T u_i \geq 0). \tag{3.8}$$

**Inference Time Model.** At the inference time, having a data sample $x \in \mathbb{R}^d$, using the $P$ vectors $u_1, \ldots, u_P$ that were used for constructing the boolean diagonal matrices $D_i$, $i \in [P]$, used in the training, the prediction is carried out similarly using the function

$$g(x, v) = \sum\nolimits_{i=1}^{P} \mathbb{1}(u_i^T x \geq 0) \cdot x^T v_i.$$

**Practical Considerations.** In experiments, we use cross-entropy loss instead of the mean square loss for the loss functions $\mathcal{L}_j$. Above, we have considered scalar output networks. In case of $k$-dimensional outputs and $k$-dimensional labels, we will simply use $k$ independent linear models parallely meaning that the overall model has a dimension $d \times P \times k$, where $d$ is the feature dimension and $P$ the number of randomly chosen hyperplanes.

### 3.4 MEETING THE REQUIREMENTS OF DP ANALYSIS

From Eq. (3.8) it is evident that each loss function $\ell(v, x_j, y_j)$, $j \in [n]$, depends only on the data entry $(x_j, y_j)$. By clipping the data sample-wise gradients $\nabla_v h(v, x_j, y_j)$, where $h(v, x_j, y_j) = \frac{1}{2} \left\| \sum_{i=1}^{P} (D_i)_{jj} x_j^T v_i - y_j \right\|_2^2$, the loss function $\ell$ becomes $2L$-sensitive (see Def. 2.7). As we explicitly show in Appendix A, the loss function $\ell(v, x_j, y_j)$ is a loss function of a generalized linear model and thus we are allowed to use the analysis of Bok et al. (2024) also when clipping the gradients since then the clipped gradients are gradients of another convex loss (Song et al., 2021). For the DP analysis, we also need to analyze the convexity properties of the loss function (3.8). We have the following Lipschitz-bound for the gradients.

**Lemma 3.2.** *The gradients of the loss function $\ell(v, x_j, y_j)$ given in Eq. (3.8) are $\beta$-Lipschitz continuous for $\beta = \|x_j\|_2^2 + \lambda$.*

Due to the $L_2$-regularization, the loss function (3.8) is clearly $\lambda$-strongly convex. The properties of $\lambda$-strong convexity and $\beta$-smoothness are preserved when clipping the sample-wise gradients $\nabla_v h(v, x_j, y_j)$ (Section E.2, Redberg et al., 2024). Thus, the DP accounting Thm. 2.8 is applicable with the same convexity parameters also when clipping the gradients $\nabla_v h(v, x_j, y_j)$.

## 4 THEORETICAL UTILITY BOUNDS IN THE RANDOM DATA MODEL

Using classical results from private empirical risk minimization (DP-ERM) we illustrate the improved convergence rate when compared to private training of 2-layer ReLU networks. In addition to having the classical convergence rate of DP-ERM, we have the approximability of ReLU networks: the minimum loss $\mathcal{L}(\theta^*, D)$ goes to zero. We emphasize that a rigorous analysis would require a priori bounds for the gradient norms. In future work, it will be interesting to see whether techniques from private linear regression (Liu et al., 2023; Avella-Medina et al., 2023; Varshney et al., 2022; Cai et al., 2021) could be used to get rid of the assumption on bounded gradients.

We consider for the problem (3.6) utility bounds with random data. This data model is also commonly used in the analysis of private linear regression (see, e.g., Varshney et al., 2022). Recently, Kim & Pilanci (2024) have given several results for convex problem (3.6) under the assumption of random data, i.e., when $X_{ij} \sim \mathcal{N}(0, 1)$ i.i.d. Their results essentially tell that taking $d$ and $n$ large enough (s.t. $n \geq d$), we have that with $P = O(\frac{n \log n/\gamma}{d})$ random hyperplane arrangements we get zero global optimum for the stochastic problem (3.6) with probability at least $1 - \gamma - \frac{1}{(2n)^8}$. If we choose $P = O\left(\frac{n \log n/\gamma}{d}\right)$ hyperplane arrangements, we have an ambient dimension $p = d \cdot P = O(n \log \frac{n}{\gamma})$ and directly get the following corollary of Thm. D.3.

**Theorem 4.1.** *Consider applying the private gradient descent (Alg. 1) to the practical stochastic strongly convex problem* (3.6) *and assume the gradients stay bounded by a constant $L > 0$. Let the ratio $c = \frac{n}{d} \geq 1$ be fixed. For any $\gamma > 0$, there exists $d_1$ such that for all $d \geq d_1$, with probability at least $1 - \gamma - \frac{1}{(2n)^8}$,*

$$\mathcal{L}(\theta^{priv}, D) \leq \widetilde{O}\left(\frac{1}{\sqrt{n}\varepsilon}\right),$$

*where $\widetilde{O}$ omits the logarithmic factors.*

## 5 DP Hyperparameter Tuning for DP-SGD and NoisyCGD

When comparing experimentally DP optimization methods, it is crucial to take into account the effect of the hyperparameter tuning on the privacy costs. The most relevant to our work are the randomized tuning methods given by Papernot & Steinke (2022) and the subsequent privacy profile-based analysis by Koskela et al. (2024). These results hold for a tuning algorithm that outputs the best model of the $K$ alternatives, where $K$ is a random variable. We consider the case $K$ is Poisson distributed, however mention that also other alternatives exist that allow adjusting the balance between compute cost of training, privacy and accuracy. The DP bounds in this case can be described as follows. Let $Q(y)$ the density function of the quality score of the base mechanism ($y$ denoting the score) and $A(y)$ the density function of the tuning algorithm that outputs the best model of the $K$ alternatives. Let $A$ and $A'$ denote the output distributions of the tuning algorithm evaluated on neighboring datasets $D$ and $D'$, respectively. Then, the hockey-stick divergence between $A$ and $A'$ can be bounded using the following result.

**Theorem 5.1** (Koskela et al. 2024)**.** *Let $K \sim \mathrm{Poisson}(m)$ for some $m \in \mathbb{N}$, and let $\delta(\varepsilon_1)$, $\varepsilon_1 \in \mathbb{R}$, define the privacy profile of the base mechanism Q. Then, for all $\varepsilon > 0$ and for all $\varepsilon_1 \geq 0$,*

$$H_{e^\varepsilon}(A||A') \leq m \cdot \delta(\widehat{\varepsilon}), \tag{5.1}$$

*where $\widehat{\varepsilon} = \varepsilon - m \cdot (e^{\varepsilon_1} - 1) - m \cdot \delta(\varepsilon_1)$.*

Theorem 5.1 says that the hyperparameter tuning algorithm is $\big(\widehat{\varepsilon}, m \cdot \delta(\widehat{\varepsilon})\big)$-DP in case the base mechanism is $(\varepsilon_1, \delta(\varepsilon_1))$-DP. If we can evaluate the privacy profile for different values of $\varepsilon$, we can also optimize the upper bound (5.1). When comparing DP-SGD and NoisyCGD, we use the fact that DP-SGD privacy profiles are approximately those of the Gaussian mechanism for large compositions (Sommer et al., 2019) as follows. Suppose DP-SGD is $(\varepsilon^*, \delta^*)$-DP for some values of the batch size $b$ noise scale $\sigma$ and number of iterations $T$. Then, we fix the value of the constant $c = \eta \cdot \lambda$ for NoisyCGD such that it is also $(\varepsilon^*, \delta^*)$-DP for the same hyperparameter values $b$, $\sigma$ and $T$, giving some GDP parameter $\mu^*$. Along the GDP privacy profile determined by $\mu^*$, we find $(\varepsilon_1, \delta(\varepsilon_1))$ that optimizes the bound of Eq. (5.1), and then evaluate the DP-SGD-$\delta$ using that same value $\varepsilon_1$, giving some value $\widehat{\delta}(\varepsilon_1)$. Taking the maximum of $\delta(\varepsilon_1)$ and $\widehat{\delta}(\varepsilon_1)$ for the evaluation of $\widehat{\varepsilon}$ in the bound of Eq. (5.1) will then give a privacy profile that bounds the DP-guarantees of the hyperparameter tuning of both DP-SGD and NoisyCGD.

## 6 Experimental Results

We compare the methods on standard benchmark image datasets: MNIST (LeCun et al., 1998), FashionMNIST (Xiao et al., 2017) and CIFAR-10 (Krizhevsky et al., 2009). The MNIST datasets have a training dataset of $6 \cdot 10^4$ samples and a test dataset of $10^4$ samples and CIFAR-10 has a training dataset size of $5 \cdot 10^4$ and a test dataset of $10^4$ samples. All samples in MNIST datasets are $28 \times 28$ size gray-level images and in CIFAR-10 $32 \times 32$ color images (with three channels each). We experimentally compare three alternatives: DP-SGD applied to a one-hidden layer ReLU network, DP-SGD applied to the stochastic convex model (3.6) without regularization ($\lambda = 0$) and NoisyCGD applied to the stochastic convex model (3.6) with $\lambda > 0$. We use the cross-entropy loss for all the models considered. In order to simplify the comparisons, we fix the batch size to 1000 for all the methods and train all methods for 400 epochs. We compare the results on two noise levels: $\sigma = 5.0$ and $\sigma = 15.0$.

Although one hidden-layer networks with tempered sigmoid activations (Papernot et al., 2021) would likely yield improved results, we focus on ReLU networks as baselines for consistency as this allows us to compare the methods in the context of ReLU-based architectures. Also, this does not affect our main finding that we are able to find improved models for the hidden state analysis.

Unlike in the experiments of, e.g., (Abadi et al., 2016), we do not use pre-trained convolutive layers for obtaining higher test accuracies in the CIFAR-10 experiment, as we experimentally observe that the DP-SGD trained logistic regression gives similar accuracies as the DP-SGD trained ReLU network. Thus, we consider the much more difficult problem of training the models from scratch using the vectorized CIFAR-10 images as features.

The hyperparameter tuning of NoisyCGD is simplified by the fact that bound of Thm. (2.8) depends monotonoysly on the parameter $c = 1 - \eta \cdot \lambda$. In case the hyperparameters $b$ noise scale $\sigma$ are

fixed, fixing the GDP parameter $\mu$ will also fix the value of $c$. Thus, if we have a grid of learning rate candidates those will also determine the values of $\lambda$'s as well. Overall, in case the batch size, number of epochs and $\sigma$ are fixed, in addition to the learning rate $\eta$, we have in all alternatives only one hyperparameter to tune: the hidden-layer width $W$ for the ReLU networks and the number of hyperplanes $P$ for the convex models. The hyperparameter grids used in the experiments are depicted in Appendix H.

We generally find that $P = 128$ is not far from the optimum for the convex model (see Appendix E.2 for comparisons using MNIST). Appendix E.1 also shows that in the non-private case, the stochastic problem approximates the ReLU problem better as $P$ increases.

## 6.1 RESULTS ON DP HYPERPARAMETER TUNING

Tables 1 and 2 show the accuracies of the best models obtained using the DP hyperparameter tuning algorithm depicted in Section 5. With the noise scale values $\sigma = 5.0$ and $\sigma = 15.0$, the DP-SGD trained models are $(1.33, 10^{-5})$-DP and $(4.76, 10^{-5})$-DP, respectively. To have similar privacy guarantees for the base models trained using NoisyCGD, we adjust the regularization constant $\lambda$ accordingly (as depicted in Secion 5) which leads to equal final $(\varepsilon, \delta)$-DP guarantees for the hyperparameter tuning algorithms. We see from tables 1 and 2 that the convex models are on par in accuracy with the ReLU network. Notice also from the results of Section 6.2 that the learning rate tuned logistic regression cannot reach similar accuracies as the NoisyCGD trained convex model.

Table 1: MNIST model accuracies vs. $\varepsilon$-values for the DP hyperparameter tuning algorithm when $\delta = 10^{-5}$. The number of candidate models $K$ is Poisson distributed with mean 20. Results are means of 5 Runs.

| $\varepsilon$ | NoisyCGD + Convex Approx. | DP-SGD + Convex Approx. | DP-SGD + ReLU |
|------|------|------|------|
| 2.88 | 0.927 | 0.929 | 0.916 |
| 8.91 | 0.944 | 0.949 | 0.944 |

Table 2: CIFAR-10 model accuracies vs. $\varepsilon$-values for the DP hyperparameter tuning algorithm when $\delta = 10^{-5}$. The number of candidate models $K$ is Poisson distributed with mean $m = 20$. Results are means of 5 Runs.

| $\varepsilon$ | NoisyCGD + Convex Approx. | DP-SGD + Convex Approx. | DP-SGD + ReLU |
|------|------|------|------|
| 2.88 | 0.416 | 0.417 | 0.427 |
| 8.91 | 0.455 | 0.459 | 0.471 |

## 6.2 RESULT WITH THE BEST MODELS

Figures 2 and 3 show the accuracies of the hyperparameter tuned models along the the training iteration of 400 epochs for the MNIST and CIFAR-10 experiment, when the privacy cost of the tuning is not taken into account. We give results for the FashionMNIST experiment in Appendix I. Figures 2 and 3 include additionally the accuracies for the learning rate optimized logistic regression models. We observe that the proposed convex model significantly outperforms logistic regression, which has been the most accurate model considered in the literature for hidden state DP analysis up to now.

Figure 2 show that the convexification helps in the MNIST experiment: both DP-SGD and the noisy cyclic mini-batch GD applied to the stochastic dual problem lead to better utility models than DP-SGD applied to the ReLU network. Notice also that the final accuracies for DP-SGD are not far from the accuracies obtained by Abadi et al. (2016) using a three-layer network for corresponding $\varepsilon$-values which can be compared using the fact that there is approximately a multiplicative difference of 2 between the two relations: the add/remove neighborhood relation used by (Abadi et al., 2016) and the substitute neighborhood relation used in our work.

Results of figures 2, 3 and I and are averaged over 5 trials and the error bars on both sides of the mean values depict 1.96 times the standard error.

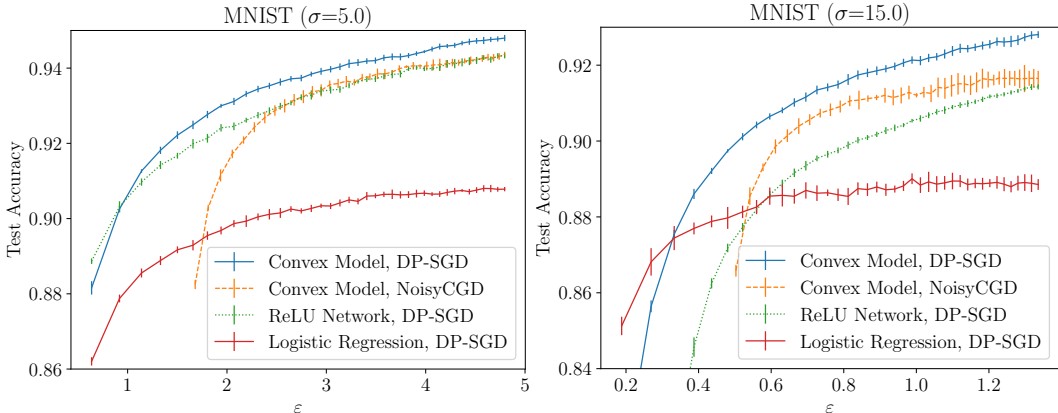

Figure 2: MNIST Comparisons: Test accuracies vs. the spent privacy budget $\varepsilon$, when $\delta = 10^{-5}$ and each model is trained for 400 epochs. The ReLU network is a one hidden-layer fully connected network, and the batch size equals 1000 for all methods considered.

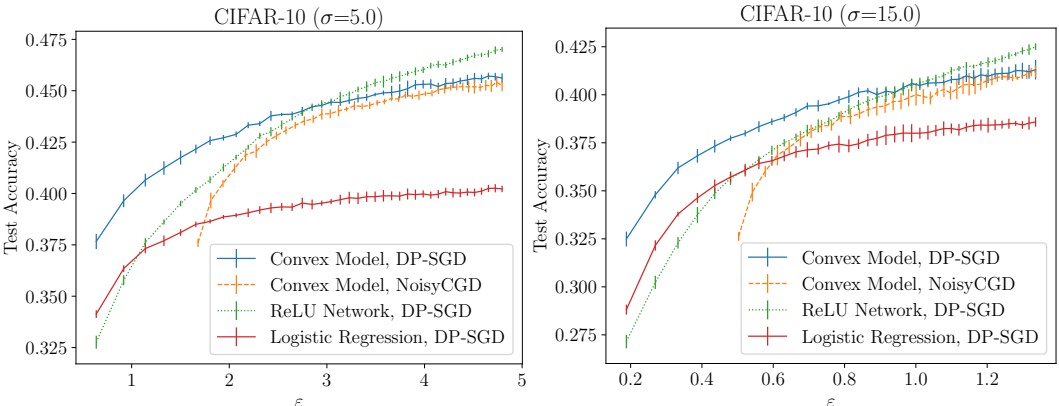

Figure 3: Cifar-10 Comparisons: Test accuracies vs. the spent privacy budget $\varepsilon$, when $\delta = 10^{-5}$ and each model is trained for 400 epochs. The model is a one hidden-layer fully connected ReLU network and the batch size equals 1000 for all methods considered.

## 7 CONCLUSIONS AND OUTLOOK

We have shown how to privately approximate the two-layer ReLU network and we have given the first high privacy-utility trade-off results using the hidden state DP analysis. In particular, we have given the first high privacy-utility trade-off results for the noisy cyclic mini-batch GD which makes it more suitable for practical applications of DP ML model training. As shown by our experiments on benchmark image classification datasets, the results for the convex problems have similar privacy-utility trade-offs as those obtained by applying DP-SGD to a one hidden-layer ReLU network and using the composition analysis. Theoretically, an interesting future task is to carry out end-to-end utility analysis for private optimization of ReLU networks via the dual form. The recent results by Kim & Pilanci (2024) could be helpful for this as they show connections between the stochastic approximation of the dual form and the ReLU minimization problem. Also, an interesting general question is, whether it is possible to obtain still better privacy-utility trade-offs for the final model in the hidden state threat model by using the privacy amplification by iteration type of analysis of, for example, DP-SGD. In order to get a better understanding of this question, tighter privacy amplification by iteration analysis for, e.g., DP-SGD or Noisy-CGD would be needed, as the composition analysis of DP-SGD cannot likely be improved a lot. Furthermore, developing DP convex models that approximate deeper neural networks (Ergen et al., 2023a), including those with convolutional layers (Ergen & Pilanci, 2020) and different activation functions (Ergen et al., 2023b), is an intriguing direction for future research.

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

## A  FORMULATING THE STRONGLY CONVEX APPROXIMATION AS A GLM

We first show that the strongly convex loss function given in Eq. (3.7) corresponds to a loss function of a convex generalized linear model. The loss function in Eq. (3.7) is of the form

$$\mathcal{L}(v, X, y) = \frac{1}{n} \sum_{j=1}^{n} \ell_j(v, x_j, y_j),$$

where

$$\ell_j(v, x_j, y_j) = \frac{1}{2} \left\| \sum_{i=1}^{P} (D_i)_{jj} x_j^T v_i - y_j \right\|_2^2 + \frac{\lambda}{2} \sum_{i=1}^{P} \|v_i\|_2^2, \quad (D_i)_{jj} = \mathbb{1}(x_j^T u_i \geq 0)$$

and $u_i$'s are the randomly sampled vectors that determine $D_i$'s (and the functions $\ell_j$) and where

$$v = \begin{bmatrix} v_1 \\ \vdots \\ v_P \end{bmatrix} \in \mathbb{R}^{P \cdot d}.$$

This is actually a generalized linear model: if we denote

$$\tilde{x}_j = \begin{bmatrix} (D_1)_{jj} x_j \\ \vdots \\ (D_P)_{jj} x_j \end{bmatrix},$$

we see that

$$\ell_j(v, x_j, y_j) = \frac{1}{2} \left\| \tilde{x}_j^T v - y_j \right\|_2^2 + \frac{\lambda}{2} \|v\|_2^2,$$

which shows that we are actually minimizing a loss function of a GLM when we are minimizing the loss $\mathcal{L}(v, X, y)$ w.r.t. $v$. By the results of (Song et al., 2021), we know that the clipped gradients are gradients of an auxiliary convex loss which allows using the privacy amplification by iteration analysis by Bok et al. (2024).

Moreover, the convexity properties of the GLM loss function are preserved under gradient clipping. This is shown in Appendix E.2 of (Redberg et al., 2024). Thus, for the privacy analysis we can use the convexity properties shown in our Section 3.4.

## B  PROOF OF LEMMA 3.2

**Lemma B.1.** *The gradients of the loss function*

$$\ell(v, x_j, y_j) = \frac{1}{2} \left\| \sum_{i=1}^{P} (D_i)_{jj} x_j^T v_i - y_j \right\|_2^2 + \frac{\lambda}{2} \sum_{i=1}^{P} \|v_i\|_2^2$$

*are $\beta$-Lipschitz continuous for $\beta = \|x_j\|_2^2 + \lambda$.*

*Proof.* For the quadratic function

$$h(v) = \frac{1}{2} \left\| \sum_{i=1}^{P} (D_i)_{jj} x_j^T v_i - y_j \right\|_2^2$$

the Hessian matrix is of the diagonal block form

$$\nabla^2 h = \text{diag}(A_1, \ldots, A_P),$$

where $A_i = x_j D_i^2 x_j^T = x_j D_i x_j^T$. Since for all $i \in [P]$, $x_j D_i x_j^T \preccurlyeq x_j x_j^T$, $\nabla^2 h \preccurlyeq x_j x_j^T$ and furthermore for the spectral norm of $\nabla^2 h$ we have that $\left\| \nabla^2 h \right\|_2 \leq \left\| x_j x_j^T \right\|_2 = \|x_j\|_2^2$.  □

## C    REFERENCE ALGORITHM FOR THE UTILITY BOUNDS

---

**Algorithm 1** Differentially Private Gradient Descent (Song et al., 2021)

---

1: Input: dataset $D = \{D_i\}_{i=1}^n$, loss function $\ell : \mathbb{R}^p \times \mathcal{X} \to \mathbb{R}$, gradient $\ell_2$-norm bound $L$, constraint set $\mathcal{C} \subseteq \mathbb{R}^p$, number of iterations $T$, noise variance $\sigma^2$, learning rate $\eta$.

2: $\theta_0 \leftarrow 0$.

3: **for** $t = 0, \ldots, T-1$ **do**

4:     $g_t^{\text{priv}} \leftarrow \frac{1}{n} \sum_{i=1}^n \partial_\theta \ell(\theta_t, d_i) + b_t$, where $b_t \sim \mathcal{N}(0, \sigma^2 I_d)$.

5:     $\theta_{t+1} \leftarrow \Pi_{\mathcal{C}} \left( \theta_t - \eta \cdot g_t^{\text{priv}} \right)$, where $\Pi_{\mathcal{C}}(v) = \text{argmin}_{\theta \in \mathcal{C}} \|\theta - v\|_2$.

6: **end for**

7: **return** $\theta^{\text{priv}} = \frac{1}{T} \sum_{t=1}^T \theta_t$.

---

## D    UTILITY BOUND WITHIN THE RANDOM DATA MODEL

We give a convergence analysis for the minimization of the loss function

$$h(v) = \frac{1}{2} \left\| \sum_{i=1}^P D_i X v_i - y \right\|_2^2$$

without any constraints. Kim & Pilanci (2024) have recently given several results for the stochastic approximations of the convex problem (3.3). The analysis uses the condition number $\kappa$ defined as

$$\kappa = \frac{\lambda_{\max}(XX^T)}{\lambda_{\min}(M)},$$

where

$$M = \mathbb{E}_{g \sim \mathcal{N}(0, I_d)}[\text{diag}[\mathbb{1}(Xg \geq 0)] X X^T \text{diag}[\mathbb{1}(Xg \geq 0)]].$$

**Lemma D.1** (Kim & Pilanci 2024, Proposition 2). *Suppose we sample $P \geq 2\kappa \log \frac{n}{\delta}$ hyperplane arrangement patterns and assume $M$ is invertible. Then, with probability at least $1 - \delta$, for any $y \in \mathbb{R}^n$, there exist $v_1, \ldots, v_P \in \mathbb{R}^d$ such that*

$$\sum_{i=1}^P D_i X v_i = y. \tag{D.1}$$

Furthermore, if we assume random data, i.e., $X_{ij} \sim \mathcal{N}(0, 1)$ i.i.d., then for sufficiently large $d$ we have the following bound for $\kappa$.

**Lemma D.2** (Kim & Pilanci 2024, Corollary 3). *Let the ratio $c = \frac{n}{d} \geq 1$ be fixed. For any $\gamma > 0$, there exists $d_1$ such that for all $d \geq d_1$ with probability at least $1 - \gamma - \frac{1}{(2n)^8}$,*

$$\kappa \leq 10\sqrt{2} \left( \sqrt{c} + 1 \right)^2.$$

There results together tell that taking $d$ and $n$ large enough (such that $n \geq d$), we have that with $P = O(\frac{n \log \frac{n}{\gamma}}{d})$ hyperplane arrangements we get the zero global optimum with high probability, i.e., there exists $u \in \mathbb{R}^{d \cdot P}$ such that Eq. (D.1) holds with probability at least $1 - \gamma - \frac{1}{(2n)^8}$.

In case $d$ and $n$ are large enough and we choose $P = O\left( \frac{n \log \frac{n}{\gamma}}{d} \right)$ hyperplane arrangements, we have that $p = d \cdot P = O(n \log \frac{n}{\gamma})$ and we directly get the following corollary.

We can directly apply the following classical result from ERM for the DP-GD (Alg. 1) to the stochastic problem (3.4) or (3.6).

**Theorem D.3** (Bassily et al. 2014; Talwar et al. 2014). *If the constraining set $\mathcal{C}$ is convex, the data sample-wise loss function $\ell(\theta, z)$ is a convex function of the parameters $\theta \in \mathbb{R}^p$, $\|\nabla_\theta \ell(\theta, z)\|_2 \leq L$ for all $\theta \in \mathcal{C}$ and $z \in D = (z_1, \dots, z_n)$, then for the objective function $\mathcal{L}(\theta, D) = \frac{1}{n} \sum_{i=1}^n \ell(\theta, z_i)$ under appropriate choices of the learning rate and the number of iterations in the gradient descent algorithm (Alg. 1), we have with probability at least $1 - \beta$,*

$$\mathcal{L}(\theta^{priv}, D) - \mathcal{L}(\theta^*, D) \leq \frac{L \|\theta_0 - \theta^*\|_2 \sqrt{p \log(1/\delta) \log(1/\beta)}}{n\varepsilon}$$

Assuming the gradients stay bounded by a constant $L$, this result gives utility bounds for the stochastic problems (3.4) or (3.6) with $p = d \times P$.

**Theorem D.4.** *Let the ratio $c = \frac{n}{d} \geq 1$ be fixed. For any $\gamma > 0$, there exists $d_1$ such that for all $d \geq d_1$, with probability at least $1 - \gamma - \frac{1}{(2n)^8}$,*

$$\mathcal{L}(\theta^{priv}, X) \leq \widetilde{O}\left(\frac{1}{\sqrt{n}\varepsilon}\right),$$

*where $\widetilde{O}$ omits logarithmic factors.*

*Proof.* By Lemma D.2, with with probability at least $1 - \gamma - \frac{1}{(2n)^8}$, for $p = d \cdot P = O(n \log \frac{n}{\gamma})$, we have that $\mathcal{L}(\theta^*, X) = 0$. Substituting this $p$ to the claim of Theorem D.3, the claim follows. $\square$

## E    ILLUSTRATIONS OF THE STOCHASTIC APPROXIMATION

### E.1    ILLUSTRATION WITH SGD APPLIED TO MNIST

Figure 4 illustrates the approximability of the stochastic approximation for the dual problem in the non-private case, when the number of random hyperplanes $P$ is varied, for the MNIST classification problem described in Section 6. We apply SGD with batch size 1000 to both the stochastic dual problem and to a fully connected ReLU network with hidden-layer width 200, and for each model optimize the learning rate using the grid $\{10^{-i/2}\}$, $i \in \mathbb{Z}$. This comparison shows that the approximabilty of the stochastic dual problem increases with increasing $P$.

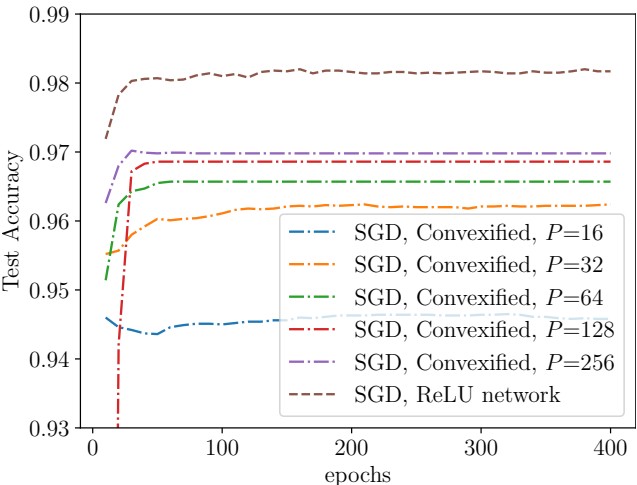

Figure 4: Test accuracies vs. number of epochs, when all models are trained using SGD with batch size 1000. The number of random hyperplanes $P$ is varied for the stochastic dual problem. The ReLU network is a one hidden-layer fully connected ReLU network with hidden-layer width 200. Cross-entropy loss is used for all models.

### E.2 ILLUSTRATION WITH DP-SGD APPLIED TO MNIST

Figure 5 illustrates the approximability of the stochastic approximation for the dual problem in the private case, when the number of random hyperplanes $P$ is varied, for the MNIST classification problem described in Section 6. We apply DP-SGD with batch size 1000 to both the stochastic dual problem and to a fully connected ReLU network with hidden-layer width 200, and for each model optimize the learning rate using the grid $\{10^{-i/2}\}$, $i \in \mathbb{Z}$. Based on these comparisons, we conclude that $P = 128$ is not far from optimum, as increasing the dimension starts means that the adverse effect of the DP noise becomes larger.

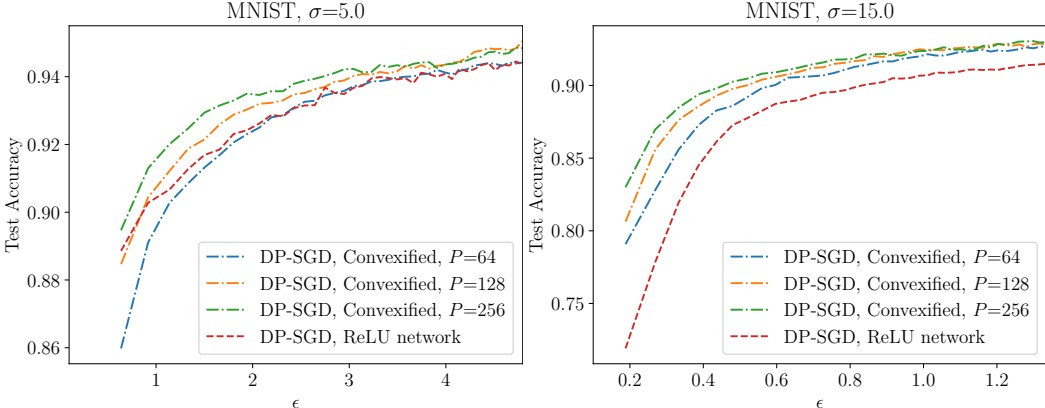

Figure 5: Test accuracies vs. number of epochs, when all models are trained using DP-SGD with batch size 1000, for two different noise levels $\sigma$. The number of random hyperplanes $P$ is varied for the stochastic dual problem. The ReLU network is a one hidden-layer fully connected ReLU network with hidden-layer width 200.

## F COMPARISON OF PLD AND RDP ACCOUNTING FOR SUBSAMPLING WITHOUT REPLACEMENT

Instead of using the numerical approach described in Section 2 of the main text, we could alternatively compute the $(\varepsilon, \delta)$-DP guarantees for DP-SGD with subsampling without replacement using the RDP bounds given by Wang et al. (2019). Fig. 6 illustrates the differences when $\sigma = 5.0$ and ratio of the batch size and total dataset size $m/n$ equals 0.01. The RDP parameters are converted to $(\varepsilon, \delta)$-bounds using Lemma F.1.

**Lemma F.1** (Canonne et al. 2020). *Suppose the mechanism $\mathcal{M}$ is $(\alpha, \epsilon')$-RDP. Then $\mathcal{M}$ is also $(\epsilon, \delta(\epsilon))$-DP for arbitrary $\epsilon \geq 0$ with*

$$\delta(\epsilon) = \frac{\exp\left((\alpha - 1)(\epsilon' - \epsilon)\right)}{\alpha} \left(1 - \frac{1}{\alpha}\right)^{\alpha-1}. \tag{F.1}$$

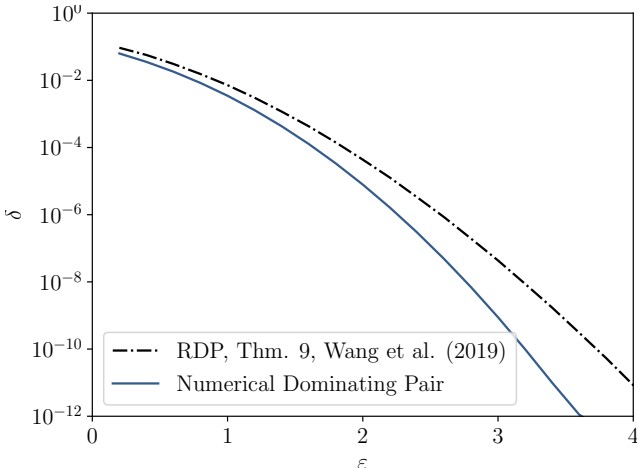

Figure 6: $(\varepsilon, \delta)$-DP guarantees for DP-SGD with subsampling without replacement, computed using the RDP bound of Wang et al. (thm. 9, 2019) and the numerical dominating pair computed using the privacy profile bound of Zhu et al. (2022) and the numerical algorithm by Doroshenko et al. (2022). The parameter $\sigma = 5.0$ and ratio of the batch size and total dataset size $m/n$ equals 0.01.

## G  REFORMULATION TOWARDS A DUAL FORM FOR THE PRACTICAL MODEL

An interesting question is whether we can interpret the loss function (3.6) suitable for DP analysis as a stochastic approximation of a dual form of some ReLU minimization problem, similarly as the stochastic problem (3.4) approximates the convex problem (3.3). We have the following result which is analogous to the reformulation behind the non-strongly convex dual form (3.3). We leave as a future work to find out whether we can state the loss function (3.6) as an approximation of some dual form.

**Theorem G.1.** *For a data-matrix $X \in \mathbb{R}^{n \times d}$, label vector $y \in \mathbb{R}^n$ and a regularization parameter $\lambda > 0$, consider the ReLU minimization problem*

$$\min_{\{u_j, \alpha_j\}_{j=1}^m} \frac{1}{2} \left\| \sum_{j=1}^m \phi(Xu_j)\alpha_j - y \right\|_2^2 + \frac{\lambda}{2} \left( \sum_{j=1}^m \|u_j\|_2^4 + \alpha_j^4 \right). \tag{G.1}$$

*Then, the problem* (G.1) *and the problem*

$$\min_{\{u_j, \alpha_j \leq 1\}_{j=1}^m, \|u_j\|_2 \forall j \in [m]} \frac{1}{2} \left\| \sum_{j=1}^m \phi(Xu_j)\alpha_j - y \right\|_2^2 + \lambda \left( \sum_{j=1}^m \alpha_j^2 \right)$$

*have equal minima.*

*Proof.* From Young's inequality $\|x\|_2^2 + \|y\|_2^2 \geq 2\langle a, b \rangle$ it follows that

$$\frac{\lambda}{2} \left( \sum_{j=1}^m \|u_j\|_2^4 + \alpha_j^4 \right) \geq \lambda \sum_{j=1}^m \|u_j\|_2^2 \cdot \alpha_j^2.$$

We see that the problem (G.1) is scaling invariant, i.e., for any solution $\{u_j^*, \alpha_j^*\}_{j=1}^m$ and for any $\gamma_i > 0, i \in [m]$, also $\{u_j^* \cdot \gamma_i, \alpha_j^*/\gamma_i\}_{j=1}^m$ gives a solution. Choosing for every $i \in [m]$, $\gamma_i = \sqrt{\frac{\alpha_i}{\|u_j\|_2}}$ gives an equality in Young's inequality. Since this scaling does not affect the solution, we must have for the global minimizer $\{u_j^*, \alpha_j^*\}_{j=1}^m$ of the ReLU minimization problem (G.1) that

$$\frac{1}{2} \left\| \sum_{j=1}^m \phi(Xu_j^*)\alpha_j^* - y \right\|_2^2 + \frac{\lambda}{2} \left( \sum_{j=1}^m \|u_j^*\|_2^4 + (\alpha_j^*)^4 \right)$$
$$= \frac{1}{2} \left\| \sum_{j=1}^m \phi(Xu_j^*)\alpha_j^* - y \right\|_2^2 + \lambda \left( \sum_{j=1}^m \|u_j^*\|_2^2 (\alpha_j^*)^2 \right). \tag{G.2}$$

Again, due to the scaling invariance, we see that the minimizing the right-hand-side of (G.2) w.r.t. $\{u_j, \alpha_j\}_{j=1}^m$ is equivalent to the problem (G.1). □

## H  HYPERPARAMETER GRIDS USED FOR THE EXPERIMENTS

The hyperparameter grids for the number of random hyperplanes $P$ for the convex model and the hidden width $W$ for the ReLU network are chosen based on the GPU memory of the available machines. For MNIST and FashionMNIST we tune the number of random hyperplanes using the grid

$$\{64, 128, 256\}$$

and for CIFAR10 using the grid

$$\{16, 32, 64\}.$$

For MNIST and FashionMNIST we tune the hidden width $W$ of the ReLU network using the grid

$$\{200, 500, 800\}$$

and for CIFAR10 using the grid

$$\{200, 400, 600\}.$$

The learning rate $\eta$ is tuned in all alternatives using the grid

$$\{10^{-3.0}, 10^{-2.5}, 10^{-2.0}, 10^{-1.5}, 10^{-1.0}, 10^{-0.5}\}.$$

## I  ADDITIONAL EXPERIMENTAL RESULTS ON FASHIONMNIST

Figure 7 shows the accuracies of the best models along the the training iteration of 400 epochs for the FashionMNISTS experiment.

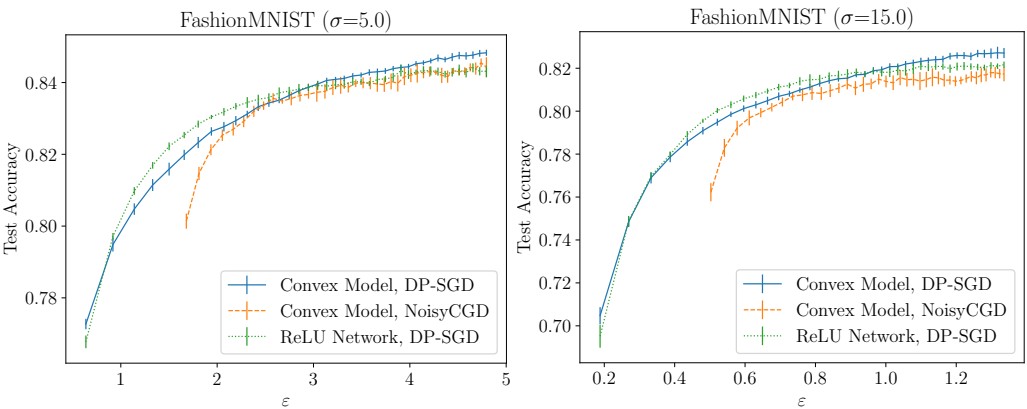

Figure 7: FashionMNIST Comparisons: Test accuracies vs. the spent privacy budget $\varepsilon$, when $\delta = 10^{-5}$ and each model is trained for 400 epochs. The model is a one hidden-layer fully connected ReLU network and the batch size equals 1000 for all methods considered.

## J  FURTHER MOTIVATION FOR NOISYCGD ANALYSIS

When using disjoint batches of data, currently the best option for obtaining rigorous guarantees is to use data shuffling and shuffling amplification (Feldman et al., 2021), however it has been shown by Chua et al. (2024a;b) that the data-shuffling combined with disjoint batches leads to an inferior privacy-utility trade-off compared to random mini-batch sampling. And we experimentally show that the method we propose (strongly convex approximation of ReLU problem + NoisyCGD) has similar privacy-utility trade-off as random mini-batch sampling applied to one hidden-layer ReLU networks. Although we do not explicitly show comparisons against the shuffled DP-SGD, we believe that our approach would be better than the shuffling approach. To illustrate this, we compute the shuffling amplification bounds by Feldman et al. (2021) by considering the setting in one of our experiments, where we use noise parameter $\sigma = 5.0$. Similarly to the experiments of Chua et al. (2024b), we use the numerical method presented in Feldman et al. (2021) to accurately compute the shuffling upper bounds. In our experiments of Section 6, we use 50 or 60 disjoint batches per

epoch. When computing the shuffling bounds, one quickly finds that this is a too few number of batches for the conditions of the analysis of Feldman et al. (2021) to hold. The shuffling privacy guarantee clearly improves the number of batches per epoch grows (see, e.g., the comparisons of Chua et al., 2024b), and to obtain a lower bound for the upper bound, we consider 1000 bathces per epoch. The comparison to the bounds of the Gaussian mechanism (i.e., no amplification) are depicted in Fig. 8. This shows that the privacy guarantees in case we use shuffling amplification bounds instead of NoisyCGD analysis in our experiments are worse than the privacy bounds of the Gaussian mechanism which further indicates that the privacy-utility trade-offs would be inferior when using data shuffling to amplify the DP guarantees.

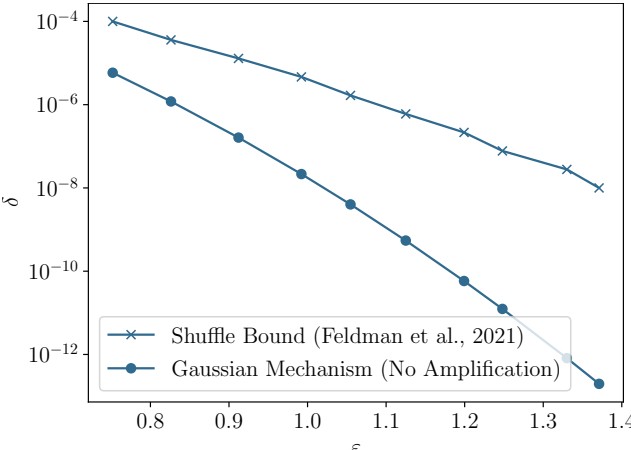

Figure 8: $(\varepsilon, \delta)$-DP guarantees for a single epoch of training when using 1000 disjoint batches and noise parameter $\sigma = 5.0$ obtained using the shuffling amplification of (Thm 3.8, Feldman et al., 2021). In experiments we use 50 or 60 batches per epoch in which case the DP guarantees of the shuffling would be even worse.

