# OpenReview forum: "Approximating Two-Layer ReLU Networks for Hidden State Analysis in Differential Privacy"
_ICLR.cc/2025/Conference — Submitted to ICLR 2025_

### Official Review · Reviewer_qA9d · 2024-10-24

**Soundness:** 3
**Presentation:** 2
**Contribution:** 3
**Rating:** 6
**Confidence:** 3

**Summary:**

This paper considers the hidden state threat model, in which the adversary doesn't have access to intermediate outputs during training. Under this model, previous works have demonstrates that the privacy cost of DP-SGD (or DP-CyclicGD) converges for convex and smooth minimization problems (no additional privacy cost for training the model for additional gradient descent steps). However, in practice, these conditions (convexity and smoothness) are not always met. A simple example of a non-convex optimization problem is the training of a two-layer ReLU network under square loss and L2 regularization.

This work focuses on addressing this issue by leveraging recent findings, which show that with square loss and L2 regularization, the 2-layer ReLU minimization problem becomes equivalent to a strongly convex minimization problem, provided the number of neurons is sufficiently large. This reformulation, being both convex and smooth, allows for the application of convergent privacy analysis under the hidden state threat model.

The authors show this DP-CGD of the alternative convex formulation enjoys similar utility-privacy tradeoff as solving the original non-convex problem with DP-SGD (under the threat model where the adversary can see all intermediate results). Specifically, they demonstrate that if the data follows a normal distribution, DP-CGD of the convex equivalence enjoys utility guarantee of the order $O(1/\sqrt{n}\epsilon)$. They also provide experimental results to support their claims.

**Strengths:**

Privacy analysis under the hidden state model is challenging for non-convex problems, as the privacy loss is typically intractable in such cases. The authors cleverly leverage recent findings that show certain non-convex problems can be reformulated as convex ones. Even though this work does not give general recipe for all non-convex problems, it is a first step towards privacy analysis under hidden state models for non-convex problems.

**Weaknesses:**

- A limitation of this work, which perhaps coincides with its advantage, is that it offers a restricted solution for privacy analysis in non-convex problems. The approach still relies on tools for convex and smooth problems, meaning that only non-convex problems that can be reformulated as convex can be analyzed. However, determining whether a non-convex problem has a convex formulation is inherently challenging.

- The preliminary section lacks coherence and is somewhat difficult to follow. Although the key concepts are introduced, their relevance to the privacy analysis is less clearly explained.

**Questions:**

1. In Figure 3, ReLU with DP-SGD outperforms DP-SGD and NosiyCGD for larger $\epsilon$. Is there any intuition for this phenomenon?

---

> ### Author Response · Authors · 2024-11-22
>
> > A limitation of this work, which perhaps coincides with its advantage, is that it offers a restricted solution for privacy analysis in non-convex problems .... determining whether a non-convex problem has a convex formulation is inherently challenging.
>
> We fully agree. Nevertheless, we would like to point out that in addition to improving the SOTA privacy-utility trade-offs in the hidden state model of DP, the proposed method (the convexified model + NoisyCGD) is actually a very competitive approach against any non-convex model in case one needs to loop over disjoint batches of data in the ML model training.
>
> When using disjoint batches of data, currently the best option is to use data shuffling and shuffling amplification, however it has been shown in [Chua et al., "How private are DP-SGD implementations?" ICML 2024](https://arxiv.org/pdf/2403.17673) that the data shuffling combined with disjoint batches leads to an inferior privacy-utility trade-off compared to random mini-batch sampling. And our method (convex model + NoisyCGD) has similar privacy-utility trade-off as random mini-batch sampling applied to one hidden-layer ReLU networks. The best privacy-utility results for NoisyCGD so far are obtained using logistic regression and we clearly improve upon that (see our experiments). Although we do not explicitly show comparisons against the shuffled DP-SGD, we believe that our approach would be better than the shuffling approach applied to any SOTA network since the privacy guarantees of the shuffled DP-SGD are so weak, see, e.g., Figure 3 in [Chua et al., 2024](https://arxiv.org/pdf/2403.17673).  Moreover, there is no accurate DP analysis for obtaining $(\varepsilon,\delta)$-DP upper bounds for shuffling of Gaussian mechanisms. The results of Figure 3 in [Chua et al., 2024](https://arxiv.org/pdf/2403.17673) are lower bounds and the actual $(\varepsilon,\delta)$-DP upper bounds proposed by
> [Feldman et al., 2023](https://arxiv.org/pdf/2208.04591) are even much worse.
>
> > In Figure 3, ReLU with DP-SGD outperforms DP-SGD and NosiyCGD for larger $\varepsilon$. Is there any intuition for this phenomenon?
>
> We believe that this is simply due to the fact that CIFAR-10 is more difficult problem than the other two datasets we consider (MNIST and FashionMNIST). Here, a larger number of hyperplanes $P$ might improve the results (we are using the fixed $P=128$ in all experiments), however the model sizes become then much bigger than the ReLU networks and also a single GPU training becomes more cumbersome with the fixed batch size of 1000 that we use for all experiments. Then again, the DP noise could start to affect more with increasing $P$. It could be that these results show the limits of this approach: the fixed random hyperplanes we use to obtain the convex model is not sufficient anymore to compete with the nonlinear ReLU network. In the non-DP literature data-adaptive hyperplanes have been used, however it is unclear how competitive that approach would be for private optimization.
>
> Nevertheless, we are improving against the SOTA results for hidden state analysis (logistic regression) in CIFAR-10 experiments as well.
>
> > The preliminary section lacks coherence and is somewhat difficult to follow. Although the key concepts are introduced, their relevance to the privacy analysis is less clearly explained.
>
> Thank you for the comment. Will restructure the preliminary section to make the relevance of the stated results clearer.

---

### Official Review · Reviewer_AHYM · 2024-10-29

**Soundness:** 2
**Presentation:** 2
**Contribution:** 3
**Rating:** 5
**Confidence:** 4

**Summary:**

The paper investigates a hidden state threat model in differential privacy, where it assumes the adversary can only access the final model. By leveraging convex reformulation of ReLU networks and privacy amplification via iterative DP analysis, the paper presents a privacy-utility trade-off in the hidden state setting.

**Strengths:**

The paper addresses an interesting problem in the differential privacy community. I am particularly intrigued by the transition from a two-layer neural network model to a convex formulation, which I have raised questions about below.

**Weaknesses:**

- The writing of the paper could be improved, as several aspects are unclear. In my opinion, the paper's structure should be: In the hidden state setting, the authors aim to achieve privacy preservation through convex approximation and privacy amplification by iteration. Specifically:
  - For the DP algorithms, the authors use NoiseCGD.
  - For convex approximation, the ReLU output is rewritten as a set of diagonal Boolean matrices, and the transition from Eq(3.2) to Eq(3.3) is made. However, I lost track in section 3.3. Several definitions and lemmas, in my opinion, should be moved to the appendix.

- There is a grammatical error in lines 270-271.

- The conclusion primarily relies on previous results.

**Questions:**

- In the hidden state setting, the authors claim that the adversary only accesses the final model. Does this mean only the final model is privatized? If not, how does this differ from the regular setting in terms of the results? Could you provide some explanations and insights?

- How should one understand the transition from Eq(3.2) to Eq(3.3)? Why does the hidden width $m$ disappear in Eq(3.3)? What is the significance of the range of $|D_x|$, even though it is related to rank, as mentioned in line 267?

- Regarding Theorem 4.1, what is $d_1$, and why is it needed? How do your results compare to DPERM with convex functions? I am confused by this result. Also, Theorem 4.1 pertains to Algorithm 1 (DP-SGD), but I am unsure if this is a typo or if I misunderstood, as I thought the authors were analyzing NoiseCGD.

---

> ### Author Response · Authors · 2024-11-22
>
> > In the hidden state setting, the authors claim that the adversary only accesses the final model. Does this mean only the final model is privatized? If not, how does this differ from the regular setting in terms of the results? Could you provide some explanations and insights?
>
> In the usual analysis of DP-SGD we use composition results and the reported $(\varepsilon,\delta)$-DP guarantees hold in case we release the whole sequence of intermediate models. When using privacy amplification by iteration - type of analysis such as that of Bok et al. (2024), the reported $(\varepsilon,\delta)$-privacy guarantees hold only in case we release a single model. The privacy amplification results then tell how the guarantees weaken as a function of number of iterations and in particular what are the $(\varepsilon,\delta)$-DP guarantees for the final model.
>
> If only we release the final model, the $(\varepsilon,\delta)$-DP guarantees we obtain using the composition analysis will hold also (releasing the final model is just post-processing of the sequence of models). So it makes sense to compare the bounds obtained using composition analysis and privacy amplification by iteration.
>
> To the best of our knowledge, DP-SGD with composition analysis has been up to now the SOTA approach in terms of privacy-utility in the hidden state model of DP. It seems that we cannot much improve upon DP-SGD either through the hidden state analysis:
> a recent work by [Annamalai, 2024](https://arxiv.org/pdf/2407.06496) shows that in the hidden state model there can't be a privacy amplification for all loss functions, when compared to the DP-SGD guarantees.
>
> However, the hidden state analysis allows analysing training methods that cannot be analysed accurately using the composition analysis. Such is for example NoisyCGD (DP-SGD iteration applied on disjoint batches of data) that has lot of practical value. When using disjoint batches of data, currently the best option is to use data shuffling and shuffling amplification results, however it has been shown in [Chua et al., "How private are DP-SGD implementations?" ICML 2024](https://arxiv.org/pdf/2403.17673) that the data shuffling combined with disjoint batches leads to an inferior privacy-utility trade-off compared to random mini-batch sampling. And our method (convex approximation + NoisyCGD) has similar privacy-utility trade-offs as random mini-batch sampling applied to one hidden-layer ReLU networks. The best privacy-utility results for NoisyCGD so far are obtained using logistic regression and we clearly improve upon that (see our experiments). Although we do not explicitly show comparisons to the shuffled DP-SGD, we believe that our approach would be better than the shuffling approach applied to SOTA models since the privacy guarantees of the shuffled DP-SGD are so weak, see, e.g., Figure 3 in [Chua et al., 2024](https://arxiv.org/pdf/2403.17673). Moreover, there is no accurate DP analysis for obtaining $(\varepsilon,\delta)$-DP upper bounds for shuffling of Gaussian mechanisms. The results of Figure 3 in [Chua et al., 2024](https://arxiv.org/pdf/2403.17673) are lower bounds and the actual $(\varepsilon,\delta)$-DP upper bounds given by Thm. 3.8 of [Feldman et al., 2021](https://arxiv.org/pdf/2012.12803) are even much worse.

---

> ### Author Response · Authors · 2024-11-22
>
> > How should one understand the transition from Eq(3.2) to Eq(3.3)? Why does the hidden width $m$ disappear in Eq(3.3)? What is the significance of the range of $|D_x|$, even though it is related to rank, as mentioned in line 267?
>
> The problems described in Eq. (3.2) and Eq. (3.3) are two different problems and they are linked via Theorem 3.1. The formulation of Eq. (3.3) lists all the possible activation patters and is a convex problem, and in case the hidden-width $m$ in Eq. (3.2) is large enough, those two problems have equal minima. Showing this equivalence is one step towards the convex approximation we consider. We will modify this part to make it more understandable.
>
> The value of $|D_x|$ will determine how many terms there are in the convex formulation of Eq. (3.3). If the rank of $X$ is high, $|D_x|$ is potentially so huge that it is not possible to solve Eq. (3.3), and thus some approximations are needed in general. To this end we consider hyperplanes determined by $P$ random vectors and formulation of Eq. (3.4).
>
> > Regarding Theorem 4.1, what is $d_1$, and why is it needed? How do your results compare to DPERM with convex functions? I am confused by this result. Also, Theorem 4.1 pertains to Algorithm 1 (DP-SGD), but I am unsure if this is a typo or if I misunderstood, as I thought the authors were analyzing NoiseCGD.
>
> First of all, we remark that there is missing an assumption from the statement of the theorem: we assume that the ratio $c = \frac{n}{d} \geq 1$ is fixed. We will add this to the statement.
>
> You are correct, we are analyzing DP-SGD applied on the convex approximation. The motivation is to illustrate the benefits of the convex approximation and to show that in addition to the convergence rate of the usual DP-ERM, we have the approximability of ReLU networks: minimum loss $\mathcal{L}(\theta^*,D)$ goes to zero.
>
> The result is essentially DP-ERM for convex functions and you get it by choosing the number of random hyperplanes $P = O(\frac{n \log n/\gamma}{d})$, where $n$ is the number of data points and $d$ is the feature dimension and $\gamma>0$.
> Then for a fixed ratio $\frac{n}{d}$, there exists a value $d_1$ such that for all $d \geq d_1$, with probability $1 - \gamma - 1/(2 n)^8$, the global minimizer of the loss function $\mathcal{L}(\theta^*,D)$ equals zero. This follows from the recent analysis of
> [Kim and Pilanci, Convex relaxations of ReLU neural networks approximate global optima in polynomial time, ICML 2024](https://arxiv.org/pdf/2402.03625v3). The total number of parameters $p = d \cdot P$, and substituting $p$ and $n$ into the classical ERM result (Theorem D.3) gives the upper bound of 4.1 for the difference $\mathcal{L}(\theta^{priv},D) - \mathcal{L}(\theta^*,D)$. As $\mathcal{L}(\theta^*,D)$ equals zero, we have something stronger than a usual DP-ERM result.
>
> In the experiments we apply DP-SGD also on the convex formulation. However,
> NoisyCGD is a separate technique that is applied on the convex formulation.  We will modify the text to make this clearer.
>
> > There is a grammatical error in lines 270-271.
>
> Thank you, we will fix this.

---

> > ### Comment · Reviewer_AHYM · 2024-11-26
> >
> > Thanks for the detailed responses. In my opinion, it’s an interesting piece of work, but it is currently under acceptance and not quite ready yet as it needs further revisions. Therefore, I will keep my score.

---

### Official Review · Reviewer_JPrt · 2024-10-31

**Soundness:** 2
**Presentation:** 2
**Contribution:** 2
**Rating:** 3
**Confidence:** 3

**Summary:**

This paper works on the hidden state threat model of differential privacy, which means the adversary only has access to the final trained machine learning model. The authors consider the model of two-layer Relu activate neural networks, which has not been studied under the hidden state threat model. The idea is to first relax the two-layer network to a stochastic convex program, then solve the relaxaztion with noisy cyclic mini-batch gradient descent. Experiment results show that the proposed method has similar performance as DP-SGD.

**Strengths:**

This paper extends the study of the hidden state threat model  to non-convex models, which is novel and interesting.

**Weaknesses:**

- The novelty seems to be limited. The major technical components, the (stochastic) convexification of 2-layer neural networks and privacy guarantee of NoisyCGD, are from previous works. This main result of this work seems to be a combination of the two components and feels incremental. Also, the paper claims that the proposed method can obtain similar privacy-utility trade-offs as DP-SGD, but actually there is no formal statement on the privacy-utility tradeoff. That is, in order to achieve $(\epsilon,\delta)$-DP in the hidden state threat model, how large the regularization parameter $\lambda$ and noise variance $\sigma^2$ we shall pick up. I do not follow how we can compare the theoretical results with DP-SGD.

- It is not clear how we shall interpret the experiment results. If I understand correctly, the hidden state threat model is a weaker threat model, because the intermediate states are not known to the adversary. So under the same privacy level, we shall need smaller noise and hence observe better utility. But the experiment result is that DP-SGD has similar or even better utility. Why is it the case? What is the advantage of introducing the hidden state threat model then?

**Questions:**

- It is hard to distinguish existing results and new finds. For instance, Theorem 5.1 appears with neither reference nor proof. If it is known, please cite where it comes from.
- In Section 3.4 you explicitly mentioned about clipping the gradients. I am not sure if this will break the privacy guarantees of NoisyCGD. [1] discussed about this point and said "...in general, clipped gradients do not correspond to gradients of a convex loss, in which case our results (as well as all other works in the literature that aim at proving convergent privacy bounds) do not apply''. Can you explain a bit more on this?

[1] Altschuler, Jason, and Kunal Talwar. "Privacy of noisy stochastic gradient descent: More iterations without more privacy loss." Advances in Neural Information Processing Systems 35 (2022): 3788-3800.

---

> ### Author Response · Authors · 2024-11-22
>
> > In Section 3.4 you explicitly mentioned about clipping the gradients. I am not sure if this will break the privacy guarantees of NoisyCGD. [1] discussed about this point and said "...in general, clipped gradients do not correspond to gradients of a convex loss, in which case our results (as well as all other works in the literature that aim at proving convergent privacy bounds) do not apply''. Can you explain a bit more on this?
>
> That same paper by [Altschuler and Talwar](https://openreview.net/pdf?id=pDUYkwrx__w) says: "For generalized linear models, the clipped gradients are gradients of an auxiliary convex loss [29], so our results can be applied directly." Here [29] refers to the paper by
> [Song et al., 2021](http://proceedings.mlr.press/v130/song21a/song21a.pdf), where the loss function corresponding to the clipped gradients is given explicitly (Section 5).
>
> In fact the strongly convex loss function of our Eq. (3.7) that we are minimizing corresponds to a loss function of a convex generalized linear model (GLM), and this can be seen as follows. The loss function in Eq. (3.7) is of the form
> $$
> \\mathcal{L}\\big(v,X,y\\big) = \\frac{1}{n} \\sum\\nolimits_{j=1}^{n} \\ell_j(v,x_j,y_j),
> $$
> where
>
> $
> \ell_j(v, x_j, y_j) = \frac{1}{2} || \sum_{i=1}^{P} (D_i)\_{jj} x_j^T v_i - y_j ||\_2^2  + \frac{\lambda}{2} \sum_{i=1}^{P} ||v_i||\_2^2,
> $
> $$
> (D_i)\_{jj}= \\mathbb{1}( x_j^T u_i \\geq 0)
> $$
> and $u_i$'s are the randomly sampled vectors that determine $D_i$'s (and the functions $\ell_j$) and where
> $$
> v = \\begin{bmatrix} v_1 \\\\ \\vdots \\\\ v_P \\end{bmatrix} \\in \\mathbb{R}^{P \\cdot d}.
> $$
> This is actually a generalized linear model: if we denote
>
> $$
> \tilde x_j = \\begin{bmatrix} (D_1)\_{jj} x_j \\\\ \vdots \\\\ (D_P)\_{jj} x_j \\end{bmatrix},
> $$
>
> we see that
> $$
> \ell_j(v,x_j,y_j) = \frac{1}{2} || \tilde x_j^T v  - y_j ||_2^2 + \frac{\lambda}{2} ||v||_2^2,
> $$
> which shows that we are actually minimizing a loss function of a GLM when we are minimizing the loss $\mathcal{L}\big(v,X,y\big)$ w.r.t. $v$.
>
> Moreover, the convexity properties of the GLM loss function are preserved under gradient clipping. This is shown in Appendix E.2 of
> [Redberg et al. (2023)](https://openreview.net/pdf?id=IpUJd3KG3c). Thus, for the privacy analysis we can use the convexity properties shown in our Section 3.4.
>
> We have already shortly mentioned the preservation of the convexity properties at the end of Section 3.4. We will modify that part and add the above shown explicit connection to GLMs to the paper (possibly Appendix).
>
> > The novelty seems to be limited. The major technical components, the (stochastic) convexification of 2-layer neural networks and privacy guarantee of NoisyCGD, are from previous works.
>
> We agree that the paper strongly relies on previous results on convexification of ReLU networks and privacy amplification by iteration. However, we think that putting things together in the right way (see, e.g., the above GLM connection that enables the privacy hidden state analysis) to actually obtain NoisyCGD-trained ML model that has similar privacy-utility tradeoffs as DP-SGD-trained one hidden layer ReLU network and to carry out careful experimental comparisons is a non-trivial task. We are improving the SOTA in privacy-utility of NoisyCGD-trained ML models and think that this also has practical value as DP-SGD with random mini-batch sampling is often hard to implement in practical scenarios.

---

> ### Author Response · Authors · 2024-11-22
>
> >  It is not clear how we shall interpret the experiment results. If I understand correctly, the hidden state threat model is a weaker threat model, because the intermediate states are not known to the adversary. So under the same privacy level, we shall need smaller noise and hence observe better utility. But the experiment result is that DP-SGD has similar or even better utility. Why is it the case? What is the advantage of introducing the hidden state threat model then?
>
> First of all, we think it makes perfect sense to compare the models in the hidden state model obtain using a) DP-SGD and composition analysis and b) NoisyCGD and privacy amplification by iteration analysis. If only we release the final model of a DP-SGD iteration, the $(\varepsilon,\delta)$-DP guarantees we obtain using the composition analysis will hold also as releasing the final model is just post-processing of the sequence of models. To the best of our knowledge, DP-SGD with composition analysis has been up to now the SOTA approach in terms of privacy-utility trade-offs in the hidden state model. You are correct, we cannot much improve upon DP-SGD either through the hidden state analysis. In fact, a recent work [Annamalai, 2024](https://arxiv.org/pdf/2407.06496) shows that in the hidden state model there can't be a privacy amplification for all loss functions, when compared to the DP-SGD guarantees. This is also our empirical observation, hidden state analysis to the convex approximation gives approximately similar privacy-utility trade-offs as DP-SGD for the final model.
>
> However, the hidden state analysis allows analysing training methods that cannot be analysed accurately using the composition analysis. Such is for example NoisyCGD (DP-SGD iteration applied on disjoint batches of data) that has lot of practical value.
> When using disjoint batches of data, currently the best option is to use data shuffling and shuffling amplification, however it has been shown in [Chua et al., "How private are DP-SGD implementations?" ICML 2024](https://arxiv.org/pdf/2403.17673) that the data shuffling combined with disjoint batches leads to an inferior privacy-utility trade-off compared to random mini-batch sampling. And our method (convex model + NoisyCGD) has similar privacy-utility trade-off as random mini-batch sampling applied to one hidden-layer ReLU networks. The best privacy-utility results for NoisyCGD so far are obtained using logistic regression and we clearly improve upon that (see our experiments). Although we do not explicitly show comparisons to the shuffled DP-SGD, we believe that our approach would be better than the shuffling approach applied to SOTA models since the privacy guarantees of the shuffled DP-SGD are so weak, see, e.g., Figure 3 in [Chua et al., 2024](https://arxiv.org/pdf/2403.17673). Moreover, there is no accurate DP analysis for obtaining $(\varepsilon,\delta)$-DP upper bounds for shuffling of Gaussian mechanisms. The results of Figure 3 in [Chua et al., 2024](https://arxiv.org/pdf/2403.17673) are lower bounds and the actual $(\varepsilon,\delta)$-DP upper bounds given by Thm. 3.8 of [Feldman et al., 2021](https://arxiv.org/pdf/2012.12803) are even much worse.
>
>
>
> > Also, the paper claims that the proposed method can obtain similar privacy-utility trade-offs as DP-SGD, but actually there is no formal statement on the privacy-utility tradeoff...
>
> You are correct, we do not make theoretical comparisons against DP-SGD applied to ReLU problem although we give a utility bound for DP-SGD applied to the convex model in random data model in our Theorem 4.1. Our main contribution is to derive a model that can be analyzed in the hidden state model of DP and which empirically leads to similar privacy-utility trade-offs as when we are applying DP-SGD to one hidden layer ReLU network and using composition results. This has the advantage that we can analyze algorithms such as NoisyCGD that cannot be analyzed accurately using compositions results.
>
> > It is hard to distinguish existing results and new finds. For instance, Theorem 5.1 appears with neither reference nor proof. If it is known, please cite where it comes from.
>
> Thank you for pointing this out, we will add the correct reference to Thm. 5.1.

---

### Official Review · Reviewer_Vyiy · 2024-11-02

**Soundness:** 3
**Presentation:** 3
**Contribution:** 2
**Rating:** 5
**Confidence:** 3

**Summary:**

This paper investigates a privacy algorithm for the two-layer ReLU minimization problem by using a convex dual formulation and privacy amplification through iterative DP analysis.

**Strengths:**

The technical approach appears largely sound (though I have not conducted a detailed verification). The core idea of transforming the two-layer ReLU network into a stochastic convex formulation to enable privacy analysis is interesting.

**Weaknesses:**

Under the restricted setting of the two-layer ReLU network, it theoretically yields an improved utility-privacy tradeoff by resorting to a dual form of the original problem. However, its applicability is limited since the approach does not extend to more general non-convex formulations. I would expect papers in this area to provide implications for broader non-convex cases, an aspect that is missing here.

**Questions:**

Can this method outperform the method of directly applying DP-SGD to a state-of-the-art (non-convex) neural network, in terms of utility-privacy tradeoff?

---

> ### Author Response · Authors · 2024-11-22
>
> > Can this method outperform the method of directly applying DP-SGD to a state-of-the-art (non-convex) neural network, in terms of utility-privacy tradeoff?
>
> Yes, in case we consider we are iterating over disjoint batches of the data (NoisyCGD). In that case, currently the best option is to use data shuffling and shuffling amplification, however it has been shown by [Chua et al., "How private are DP-SGD implementations?" ICML 2024](https://arxiv.org/pdf/2403.17673) that the data shuffling combined with disjoint batches leads to an inferior privacy-utility trade-off compared to random mini-batch sampling. And our method (convex model + NoisyCGD) has similar privacy-utility trade-off as random mini-batch sampling applied to one hidden-layer ReLU networks. The best results in terms of privacy-utility for NoisyCGD so far are obtained using logistic regression and we clearly improve upon that (see our experiments). Although we do not explicitly show comparisons to the shuffled DP-SGD, we believe that our approach would be competitive compared to the shuffling approach applied to SOTA networks since the privacy guarantees of the shuffled DP-SGD are so weak, see, e.g., Figure 3 in [Chua et al., 2024](https://arxiv.org/pdf/2403.17673). Notice also that there is no accurate DP analysis for obtaining $(\varepsilon,\delta)$-DP upper bounds for shuffling of Gaussian mechanisms. The results of Figure 3 in [Chua et al., 2024](https://arxiv.org/pdf/2403.17673) are lower bounds and the actual $(\varepsilon,\delta)$-DP upper bounds given by Thm. 3.8 of [Feldman et al., 2021](https://arxiv.org/pdf/2012.12803) are even much worse.
>
> > I would expect papers in this area to provide implications for broader non-convex cases, an aspect that is missing here.
>
> Thank you for the comment. It is an interesting question whether this method can be generalized to other activation functions and to multiple layers. In the non-DP literature this has already been done, e.g., for networks with threshold activation functions ([Ergen, Tolga, et al. "Globally Optimal Training of Neural Networks with Threshold Activation Functions." ICLR 2023](https://openreview.net/pdf?id=_9k5kTgyHT)  and for multiple layer networks ([Ergen and Pilanci. "Revealing the structure of deep neural networks via convex duality."ICML 2021](https://arxiv.org/pdf/2002.09773)). It looks like those model could be trained privately. We have added these references as pointers to possible future work in the conclusions section.
>
> Nevertheless, we think that the privacy-utility gap between the previously considered models for hidden state analysis such as logistic regression and the convex approximation of the ReLU network we consider is so big that this method has value already as such. And we think that the effort to put pieces together to obtain NoisyCGD-trained models with this good privacy-utility tradeoffs is a non-trivial effort. We think is that this would be a very competitive approach in case one has to carry out DP-SGD on fixed disjoint batches of data, something that is often needed in practice.

---

### Author Response · Authors · 2024-11-26
**General Rebuttal**

We would like to thank all the reviewers for their constructive comments!

Below, we have addressed most of the questions and comments individually. We have also uploaded an updated version of the paper with two major modifications:

* Adding a section to the appendix (Appendix A). We explicitly demonstrate that the strongly convex loss function we derive corresponds to the loss function of a GLM. Together with the results of Song et al. (2021), this shows the existence of another convex loss function whose gradients match the clipped gradients and that the privacy amplification by iteration analysis is applicable. This addresses the concern raised by reviewer JPrt regarding the validity of the privacy amplification by iteration analysis when clipping gradients.

* Adding a section in the appendix (Appendix J). We illustrate that, in our experimental settings, the state-of-the-art shuffling amplification upper bounds are worse than the privacy bounds of the Gaussian mechanism. This indicates that the results we obtain with NoisyCGD push the state-of-the-art in the privacy-utility trade-off of DP-SGD with disjoint batches. Adding this section is in response to reviewers who questioned the need for convex approximation and hidden state analysis when DP-SGD with the non-convex ReLU problem is competitive. As we also highlight in invidual responses, this is because it facilitates the analysis of other practically relevant algorithms, such as DP-SGD with disjoint batches (NoisyCGD).

We have also made minor modifications based on the reviewers' feedback. These include adding references for future work in the conclusions section and clarifying that the theoretical utility bound, in addition to providing a classical DP-ERM convergence rate, incorporates the approximability of ReLU networks as the minimum loss in the bound approaches zero.

---

### Meta-Review · Area_Chair_Fg2e · 2024-12-20

**Metareview:**

The reviewers generally agree that the paper has some merits but falls short of the acceptance threshold.  The main concerns are the limited novelty, lack of generalizability to broader non-convex cases, and unclear practical advantages compared to DP-SGD.

Specifically, reviewers question whether the method can outperform DP-SGD in terms of utility-privacy tradeoff and whether it has broader implications beyond the two-layer ReLU network.  The authors' response, while acknowledging these limitations, emphasizes the practical value of their approach for DP-SGD with fixed disjoint batches of data (NoisyCGD), which is common in practice.  However, the reviewers seem unconvinced and still find the novelty and contribution insufficient for acceptance.

**Additional Comments On Reviewer Discussion:**

See above.

---

### Decision · Program_Chairs · 2025-01-22

Reject